# GRADIENT-INFORMED QUALITY DIVERSITY FOR THE ILLUMINATION OF DISCRETE SPACES

## ABSTRACT

Quality Diversity (QD) algorithms have been proposed to search for a large collection of both diverse and high-performing solutions instead of a single set of local optima. While early QD algorithms view the objective and descriptor functions as black-box functions, novel tools have been introduced to use gradient information to accelerate the search and improve overall performance of those algorithms over continuous input spaces. However a broad range of applications involve discrete spaces, such as drug discovery or image generation. Exploring those spaces is challenging as they are combinatorially large and gradients cannot be used in the same manner as in continuous spaces. We introduce MAP-ELITES with a Gradient-Informed Discrete Emitter (ME-GIDE), which extends QD optimisation with differentiable functions over discrete search spaces. ME-GIDE leverages the gradient information of the objective and descriptor functions with respect to its discrete inputs to propose gradient-informed updates that guide the search towards a diverse set of high quality solutions. We evaluate our method on challenging benchmarks including protein design and discrete latent space illumination and find that our method outperforms state-of-the-art QD algorithms in all benchmarks.

## 1 INTRODUCTION

Quality-Diversity (QD) Optimization algorithms Chatzilygeroudis et al. (2021); Cully & Demiris (2017) have changed the classical paradigm of optimization: inspired by natural evolution, the essence of QD methods is to provide a large and diverse set of high-performing solutions rather than only the best one. This core idea showed great outcomes in different fields such as robotics Cully et al. (2015) where it allows to learn to control robots using diverse and efficient policies or latent space exploration Fontaine et al. (2021) of generative models to generate a diverse set of high quality images or video game levels. The main QD approaches are derivative-free optimizers, such as the MAP-ELITES algorithm Mouret & Clune (2015), a genetic algorithm that tries to find high quality solutions covering the space defined by a variety of user-defined features of interest. However, subsequent methods Nilsson & Cully (2021); Fontaine & Nikolaidis (2021) have shown that when the objective functions (quality) or the descriptor functions (diversity) are differentiable, using gradients of those functions can improve convergence both in terms of speed and performance compared to traditional QD algorithms. Those methods were applied to Reinforcement Learning problems such as robotics Pierrot et al. (2022a) or latent space illumination Fontaine & Nikolaidis (2021) of generative adversarial networks.

Those applications focused on optimization over continuous variables. However many real-world applications are best framed using discrete features. For instance, generative architecture such as discrete VAEs Van Den Oord et al. (2017); Rolfe (2016) have shown capability to generate high-quality images Razavi et al. (2019) and have been at the core of recent successful approaches for text-to-image generation Ramesh et al. (2021); Gafni et al. (2022). A further natural application using discrete variables is protein design Huang et al. (2016): proteins play a role in key functions in nature and protein design allows to create new drugs or biofuels. As proteins are sequences of 20 possible amino acids, designing them is a tremendously hard task as only a few of those possible sequences are plausible in the real world. Deep learning for biological sequences provided significant advances in several essential tasks for biological sequence, such AlphaFold for structure prediction Jumper et al. (2021a) or language models for amino acid likelihood prediction Rives et al.

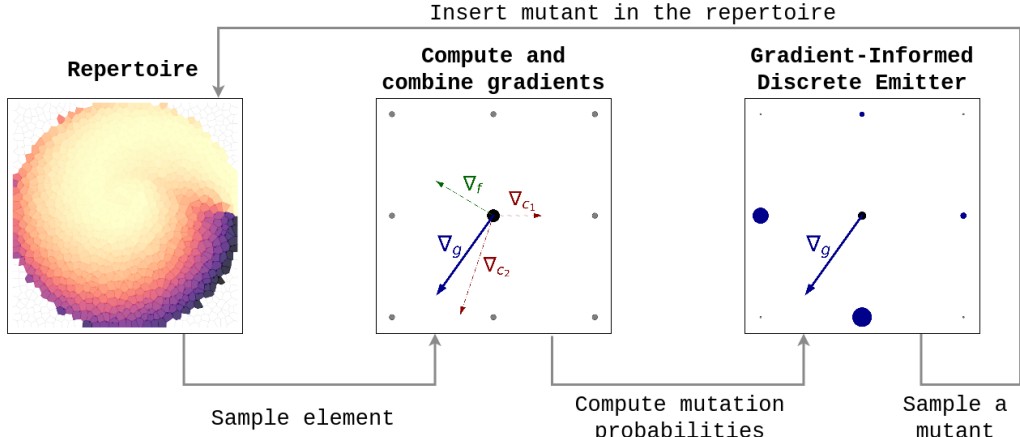

Figure 1: MAP-ELITES with Gradients Informed Discrete Emitter (ME-GIDE). At each iteration, an element is sampled in the repertoire. Gradients are computed over continuous fitness and descriptor functions with respect to their discrete inputs. Gradients are linearly combined to favour higher fitness and exploration of the descriptor space. Probabilities of mutation over the neighbours of the element are derived from this gradient information. Finally, a mutant is sampled according to those probabilities and inserted back in the repertoire.

(2021) . These successes motivate the use of these tools as objective functions for protein design Anishchenko et al. (2021a).

In practice, objective functions are often written as differentiable functions of their inputs, enabling the use of gradient information of the continuous extension if the input are discrete Grathwohl et al. (2021). We second this research trend and address gradient informed quality-diversity optimization over discrete inputs when objective and descriptors functions are differentiable. We make several contributions: (i) we introduce MAP-ELITES with a Gradient-Informed Discrete Emitter (ME-GIDE), a genetic algorithm where mutations are sampled thanks to a gradient informed distribution. (ii) We propose a way to select the hyperparameters of this algorithm by weighting the importance we give to this gradient information. (iii) We demonstrate the ability of our method to better illuminate discrete spaces on proteins and the discrete latent space of a generative model.

## 2 PROBLEM DEFINITION

In this work, we consider the problem of Quality Diversity Optimization over discrete spaces. The QD problem assumes an objective function $f : \mathcal{X} \to \mathbb{R}$, where $\mathcal{X}$ is called search space, and $d$ descriptors $c_i : \mathcal{X} \to \mathbb{R}$, or as a single descriptor function $\mathbf{c} : \mathcal{X} \to \mathbb{R}^d$. We note $S = \mathbf{c}(\mathcal{X})$ the descriptor space formed by the range of $\mathbf{c}$. We only consider discrete spaces of dimension $m$ with $K$ categories such that $\mathcal{X} \subset \{1, ..., K\}^m$. QD algorithms of the family of the MAP-ELITES algorithm, discretize the descriptor space $S$ via a tessellation method. Let $\mathcal{T}$ be the tessellation of $S$ into $M$ cells $S_i$. The goal of QD methods is to find a set of solutions $\mathbf{x}_i \in \mathcal{X}$ so that each solution $\mathbf{x}_i$ occupies a different cell $S_i$ in $\mathcal{T}$ and maximizes the objective function within that cell. The QD objective can thus be formalized as follows:

$$\max_{\mathbf{x} \in \mathcal{X}} \sum_{i=1}^{M} f(\mathbf{x}_i), \text{ where } \forall i, \ \mathbf{c}(\mathbf{x}_i) \in S_i. \tag{1}$$

We consider here that both the objective and descriptor functions are actually defined on real values $\mathbb{R}$ and restricted to the discrete inputs $\mathcal{X}$. We also consider them to be first-order differentiable, hence for any input $x \in \mathcal{X}$, we can compute gradients $\nabla_x f(x)$ and $\nabla_x c_i(x)$.

---

**Algorithm 1:** Gradient Informed Discrete Emitter (GIDE)

---

**Given:** a batch of $B$ elements $x_n$ from the repertoire.

**for** $1 \leq n \leq B$ **do**

    // Compute the distribution

    Compute $\tilde{\mathbf{d}}_\mathbf{n}$ using Equation 2.

    Compute the proposal distribution $p_n$ using Equation 4.

    // Target entropy

    Adjust $T$ to obtain a target entropy $\bar{H} = \bar{H}_{\text{target}}$ over the proposal distributions.

    // Sample

    Draw a mutant $x'_n$ from each $x_n$ according to $p_n$.

**end**

Return $x'_1, ..., x'_B$

---

## 3 MAP-Elites with Gradient Proposal Emitter

**MAP-Elites and Differentiable MAP-Elites** MAP-ELITES is a QD method that discretizes the space of possible descriptors into a repertoire (also called archive) via a tessellation method. Its goal is to fill each cell of this repertoire with the highest performing individuals. To do so, MAP-ELITES firstly initializes a repertoire over the BD space and secondly initializes random solutions, evaluates them and inserts them in the repertoire. Then successive iterations are performed: (i) select and copy solutions uniformly over the repertoire (ii) mutate the copies to create new solution candidates (iii) evaluate the candidates to determine their descriptor and objective value (iv) find the corresponding cells in the tessellation (v) if the cell is empty, introduce the candidate and otherwise replace the solution already in the cell by the new candidate if it has a greater objective function value. When using real variables, the most popular choice is the Iso+LineDD Vassiliades & Mouret (2018) that mixes two gaussian perturbations. When using discrete variables, mutations are generally defined as point mutation, *ie* select a random position and flip its value from the current one to a new one.

To incorporate gradients into MAP-ELITES algorithms, (Fontaine & Nikolaidis, 2021) introduced MAP-ELITES with a Gradient Arborescence (MEGA) a novel way to use gradient information to guide the mutations. First, the authors propose a novel objective function to encompass both quality and diversity: $g(x) = f(x) + \sum_{i=1}^{d} w_i c_i(x)$, where $w_i \sim \mathcal{N}(0, \sigma_g I) \, \forall i \in \{0, ..., d\}$. In one version of their algorithm, OMG-MEGA, authors simply extend MAP-ELITES by mutating selected element of the repertoires $x$ via $x' = x + |w_0| \nabla_x g(x)$ where $w_i$ are sampled at each iteration and for each element in the batch and $\sigma_g$ acts similarly to a learning rate. Indeed, maximizing $g$ will direct the mutations towards higher fitness and different directions of the descriptor space thanks to the randomness introduced by sampling the $w_i$.

**From Gradients to Gradient Informed Discrete Emitter** As this approach has proven effective on several tasks, we follow the previous formulation by trying to maximize $g$. In this case at a given iteration, for a given sampled $x \in \mathcal{X}$ and given sampled coefficients $c_{0:...:d}$, our mutation should ideally find a neighbour $x' \in \mathcal{X}$ that maximizes $g$. Ideally, our operator should provide $x' = \arg\max_{z \in H_\tau(x)} g(z) - g(x)$. where $H_\tau(x)$ the Hamming ball of size $\tau$ around $x$ our mutation.

However, for a given variable $x \in \{1, ..., K\}^m$, even for $\tau = 1$ the cardinality of this Hamming ball is $mK - 1$. Hence finding the optimal $x'$ requires $mK - 1$ evaluations of $g$. Doing it at each step would be too expensive but since those differences are local, they can be approximated using gradient information. Following Grathwohl et al. (2021) we use Taylor-series approximation of the local differences: noting $x^{(i,k)}$ the mutant from $x$ by flipping the position $i$ from $x_i$ to $k$, we estimate those differences as

$$g(x^{(i,k)}) - g(x) \simeq \nabla_x g(x)_{ik} - x_i^T \nabla_x g(x)_i = \tilde{d}_{ik} \tag{2}$$

where $x'$ differs from $x$ by a flip on position $1 \leq i \leq m$ from $x_i$ to $k \in \{1, ..., K\}$. The previous formulation works in the case where $\tau = 1$ but similar approximations can be derived for larger window sizes at the expense of the approximation's quality.

Using those approximations, we could straightforwardly compute a local maximum of $g$ by taking the argmax at each step. In order to encourage exploration, we use those approximate local differences to create a proposal distribution over the neighbours of $x$. Inspired by sampling techniques such as Metropolis-Hastings, we define the probability $p(x^{(i,k)}|x)$ of mutating from $x$ to $x^{(i,k)}$ as

$$p(x^{(i,k)}|x) \propto e^{\frac{\tilde{d}_{ik}}{T}} \tag{3}$$

where $T > 0$ is a temperature parameter. The temperature parameter tempers whether the mutation will go towards the argmax ($T \to 0$) or a purely random mutation ($T \to +\infty$). Finally, in order to sample flips at position $i$ from $x_i$ to $k$, we normalize those probabilities by computing a softmax over every possible flip ($mK$ possibilities):

$$p(x^{(ik)}|x) = \frac{e^{\tilde{d}_{ik}/T}}{\sum\limits_{i,l=1}^{m,K} e^{\tilde{d}_{jl}/T}} \tag{4}$$

As every approximated difference $\tilde{d}_{ik}$ can be computed in one gradient evaluation $\nabla_x g(x)$, those probabilities can be efficiently computed.

Formally our Gradient Informed Discrete Emitter (GIDE) receives a candidate $x$, computes $\tilde{d}_{ik}$ using Equation 2, then computes mutation probabilities using Equation 4 and finally samples a mutated $x^{(i,k)}$ using these probabilities. We summarize this procedure in Algorithm 1.

Using this emitter, we design our main algorithm: MAP-ELITES with a Gradient Informed Discrete Emitter (ME-GIDE). The procedure follows the one of MAP-ELITES: we firstly initialize a repertoire of behaviour descriptors. Then at each iteration, the following operations are performed: (i) select a batch of solutions uniformly over the repertoire (ii) sample mutants using our GIDE (iii) evaluate the candidates and add them in the repertoire if their fitness is higher than the existing solution in the cell. This procedure is described in Algorithm 2 with an additional step to control the strength of the gradient guidance defined in the following paragraph.

---

**Algorithm 2:** MAP-Elites with Gradient Informed Discrete Emitter (ME-GIDE)

**Given:** the number of cells $M$ and tesselation $\mathcal{T}$ ; the batch size $B$ and the number of iterations $N$ ;the descriptors function $\mathbf{c}$ and the multi-objective function $\mathbf{f}$ ; a target entropy value $H_{target}$ ; the initial population of solution candidates $\{\mathbf{x}_k\}$

```
// Initialization
```
For each initial solution, find the cell corresponding to its descriptor and add initial solutions to their cells.

```
// Main loop
```
**for** $1 \leq n_{steps} \leq N$ **do**

    ```// Select new generation```
    Sample uniformly $B$ solutions $x_n$ in the repertoire with replacement

    ```// Compute the gradients```
    Randomly draw weights $\mathbf{w^{(n)}} \sim \mathcal{N}(0, I)$
    Compute gradients $\nabla_{x_n} f(x_n)$ and $\nabla_{x_n} c(x_n)$
    Normalize the gradients.

    Compute combined updates $\nabla_n = |w_0^{(n)}| \nabla_{x_n} f(x_n) + \sum\limits_{i=1}^{n} w_i^{(n)} \nabla_{x_n} c_i(x_n)$

    ```// Gradient-Informed Discrete Emitter```
    Use Algorithm 1 to sample mutants $x'_n$

    ```// Addition in the archive```
    Add each mutant in its corresponding cell if it improves the cell fitness, otherwise discard it.

**end**

---

**Controlling the Gradient Guidance with a Target Entropy**   The shape of the flip distribution $p_{i,k}$ computed by GIDE strongly depends on the choice of the temperature parameter $T$. However, searching for an optimal value of this unbound parameter can be tedious owing to the fact that it lies in $[0, +\infty]$ and has limited human-interpretability. Since this temperature parameter $T$ controls the concentration of the probability mass over the directions with highest estimated-improvement values $\tilde{d}_{i,k}$, mispecifying it would result in suffering either degenerated or uniform distribution. The former means being too greedy towards the most promising directions, which hinders exploration, while the latter means under-exploiting our estimate. We propose to tackle this exploration-exploitation tradeoff by controlling the Shannon entropy of the flip-distribution $H(p_{i,k})$. When $T \to 0$, we approach the deterministic distribution over the highest estimate, i.e. with minimum entropy and when $T \to +\infty$ we approach the uniform distribution, i.e. with maximum entropy.

To avoid either degenerate cases, we set a target $\bar{H}_{\text{target}}$ for the normalized entropy $\bar{H}(p_{i,k}) = \frac{H(p_{i,k})}{-\log(nm)} \in [0, 1]$ of the flip distribution. We aim to find a value $T$ such that $\bar{H}(p_{i,k}) = \bar{H}_{\text{target}}$. The relationship between the normalized entropy and the temperature is non-trivial and depends on the gradients of $g$. Using Equation 4, we can write the normalized entropy equation as a function of the temperature T, we solve numerically for $T$ by minimizing by gradient descent the penalty. In practice, to select an appropriate target $\bar{H}_{\text{target}}$, we find it enough to test a few values in $\bar{H}_{\text{target}} \in \{0.4, 0.6, 0.8\}$. One can note that setting $\bar{H}_{\text{target}} = 1$ corresponds to completely random mutations.

## 4    EXPERIMENTS

### 4.1    SETTINGS

We conduct experiments on different benchmarks to assess the performance of our new emitter. Namely, we experiment on three challenging applications: illuminating a family of proteins, generating diverse images by working directly on the discretized pixel space and illuminating the discrete latent space of a generative model.

**Design in the Ras Protein Family**   As proteins are involved in numerous biological phenomena, including every task of cellular life, designing them has been a longstanding issue for biologists. The simplest way to describe a protein is through its primary structure: a natural protein is a sequence of 20 basic chemical elements called amino-acids. Designing a protein is generally framed as finding the sequence of amino acids that maximizes a score representing a property of a protein, such as its ability of binding to a virus or its stability. It is a tedious task as the search space grows exponentially with the length of the protein of interest, for instance designing a protein of 100 residues operates over a space of $20^{100}$ elements. In our experiment, we aim to redesign members of the "Rat sarcoma virus" (*Ras*) protein family (extracted from PFAM database (Mistry et al., 2021)), involved in cellular signal transduction and plays a role in cancer development Bos (1989).Following a common procedure in computational protein design Huang et al. (2016), we aim to redesign its members to maximize their stability. While many methods use biophysical simulations to estimate this stability Desmet et al. (1992), we follow recent trends Anishchenko et al. (2021b) and use a pre-trained neural network to approximate it.

Namely, for a given sequence, our objective function is defined as the log-likelihood provided by a pre-trained protein language model: ESM2 Lin et al. (2022); Rives et al. (2021). Indeed, it was shown that this quantity is correlated to the stability of a protein and can even be used to predict the effect of these mutations Hopf et al. (2017); Riesselman et al. (2018). We use an unsupervised procedure to create descriptors similar to AURORA Grillotti & Cully (2022): we sample 1,000,000 proteins from the Uniref50 [1] database, compute their 640-dimensional embeddings using ESM2 at the $20^{th}$ layer and perform a PCA on those embeddings to extract 5 components. Finally we use a $K$-Means algorithm with $K = 30000$ on these projected embeddings to create the tesselation with $30,000$ cells. The descriptors of a sequence is the corresponding projected embedding of this sequence and is mapped to its corresponding cell by finding the closest centroid. In practice, we use the $120,000$ elements of this family at initialization.

---

[1]`https://www.uniprot.org/help/uniref`

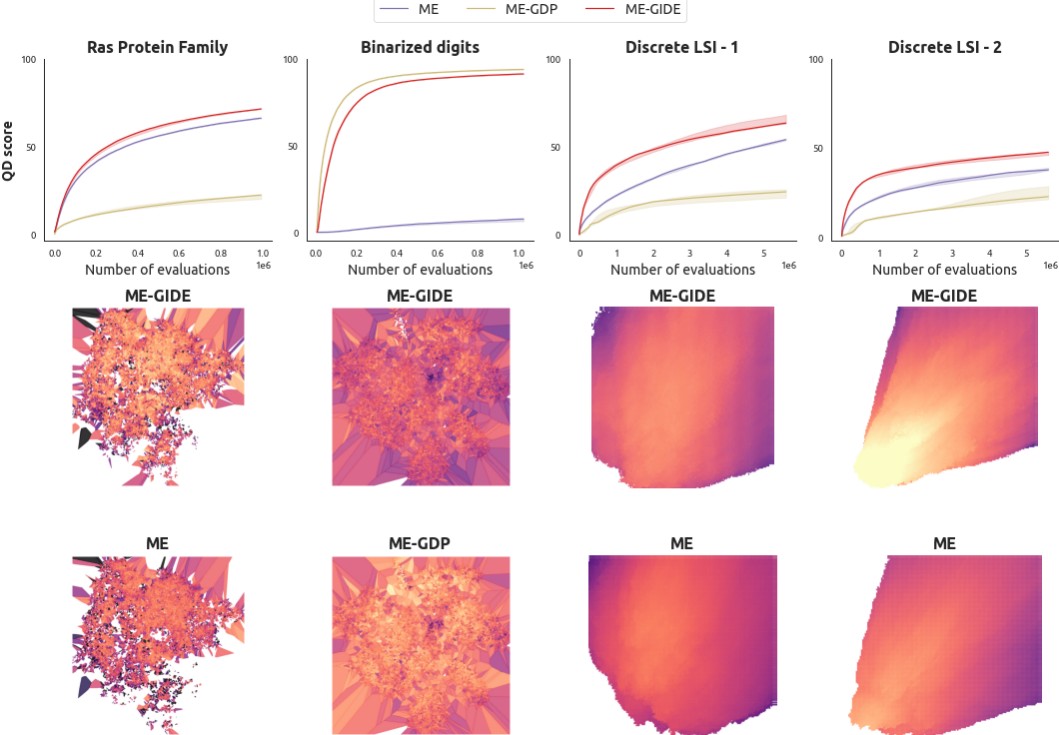

Figure 2: **(Top)** QD-score evolution on the different domains (median and interquartile range over 5 seeds). ME-GIDE outperforms MAP-ELITES on every experiment and ME-GDP on every experiment except for Binarized Digits. Only the solution corresponding to the best set of hyperparameters are shown for MAP-ELITES and ME-GDP. **(Middle)** Final repertoire of ME-GIDE for one seed. A yellow color corresponds to a high fitness, whereas darker colours correspond to lower fitnesses. **(Bottom)** Repertoire of the best performing baseline.

**Binarized Digits** To further illustrate the interest of our method, we design an experiment on binary data consisting in generating diverse MNIST digits with QD methods. Given our fitness and descriptor functions, we aim to find out how QD methods compare in generating diverse digits by directly searching over the image space. We define the fitness using an energy-based model trained on discrete MNIST data. Following Grathwohl et al. (2021), we train a restricted Boltzmann machine (RBM) Hinton (2012) on the binarized 28x28 images. To obtain descriptors we define a diversity space by embedding the MNIST dataset into the hidden layer of the RBM and then by computing a PCA over these embeddings. Thus, descriptors are defined as projections over the top-$d$ components of this PCA with $d = 20$. We again finally use a $K$-Means algorithm with $K = 10000$ on these projected embeddings to create the tesselation with $10,000$ cells. We display in Appendix D how this descriptor space spans the different digits' classes. In our experiments, we initialize our images with a randomly uniform distribution.

**Discrete Latent Space Illumination (LSI)** Recent works Fontaine et al. (2021), Fontaine & Nikolaidis (2021) have proposed the latent space illumination of generative models (LSI) problem as a benchmark for QD methods. It consists in searching the latent space of a large generative model for diverse and relevant latent codes. The diversity of the sampled images being a major concern for generative models, it perfectly aligns with the QD methods design. For instance, Fontaine & Nikolaidis (2021) uses QD methods to illuminate the latent space of a StyleGAN, they leverage CLIP Radford et al. (2021) prompts to create both a fitness function and a descriptor space and to generate diverse images that satisfy some prompts. Since our work is focused on discrete variables, we instead chose to illuminate the latent space of a discrete VAE. Specifically, we choose to explore the latent space of a vector-quantized auto-encoder (VQ-VAE). Similarly to previous work, we use CLIP to evaluate the quality of the VQ-VAE-decoded latents as well as their diversity. We have experimented two sets of prompts. The first one aims to generate labrador dog images, the fitness is

defined with the prompt "A labrador" while the descriptor functions are defined as "A white puppy" and "A dog with long hair". The second set of prompt searches for truck images with fitness prompt "A truck" and descriptor prompts "A red truck", "A blue truck". The aim of this experiment is to find a large set of latent codes which - once decoded - yield diverse images with high scores according to the CLIP evaluation. We refer the reader to the Appendix C for the VQ-VAE model's details, and the training procedure. We want to stress that our goal is only to explore the latent space of the VQ-VAE in the light of the CLIP score and descriptors functions, we do not aim to compete with image generation procedure nor with text-to-image approaches.

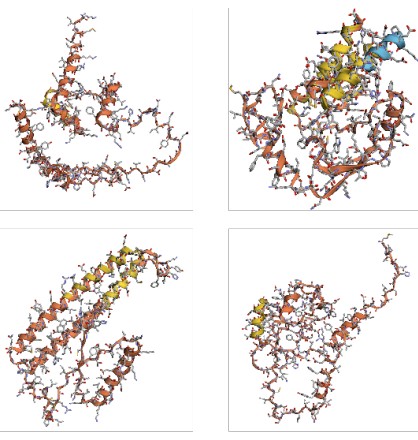

Figure 3: *Ras* proteins designed by ME-GIDE, visualized using the prediction of AlphaFold. One can see diverse patterns in the secondary structure, while still retaining high estimated stability.

**Experiment Setup** In addition to our ME-GIDE algorithm, we compare with two main baselines. The first baseline, MAP-ELITES stands for the original MAP-ELITES algorithm with completely random mutations at each step. Searching via random mutations is the most popular solution when no prior knowledge is assumed over the structure of a solution. We also consider additional variation operators such as crossover. We also compare ME-GIDE to a natural baseline: gradient descent with projection (ME-GDP). Inspired by Fast SeqProp Linder & Seelig (2020), ME-GDP uses gradients following Objective and Measure Gradient MAP-ELITES via Gradient Arborescence (OMG-MEGA) Fontaine & Nikolaidis (2021) and projects the mutants back to the discrete set. We implement every method based on Jax using the QDax open source library Lim et al. (2022).

We run every experiment over $5$ different seeds for each method. We use a repertoire with $30,000$ centroids for the protein experiment, $10,000$ centroids for the binarized digits experiments and $40,000$ centroids for the discrete LSI experiments. We stop the illumination after evaluating $1e6$ candidates on the protein and binarized digits experiment and $5e6$ evaluations for the discrete LSI experiment. We provide more details about the settings in Appendix A.

## 4.2 RESULTS AND ANALYSIS

Figure 2 summarizes the results of our experiments, we display the median and the interquartile range over 5 seeds for every benchmark. In the Ras and Discrete LSI experiments, ME-GIDE outperforms other methods, since it converges in fewer iteration to higher performances. For clarity, we only plot the results corresponding for the best set of hyperparameters for the baselines MAP-ELITES and ME-GDP. We use a Wilcoxon signed-rank test Wilcoxon (1992) to compare the distribution of the scores obtained over different seeds where the null hypothesis is that the obtained scores have the same median. It shows that ME-GIDE constantly outperforms MAP-ELITES in the QD score with p-values lower than $0.05$ in every experiment and outperforms ME-GDP on every experiment except Binarized Digits.

Our target entropy procedure allows an easier choice of hyperparameters for ME-GIDE. Indeed, in our experiment, we only try for three values: $0.4$, $0.6$ and $0.8$. While we display the best results for clarity on Figure 2, the conclusion are similar for values in the range $[0.4, 0.8]$ as detailed in Appendix B.

Interestingly, ME-GDP under-performs compared to purely random mutations for any set of hyperparameters on both protein and VQ-VAE experiments. This means performing gradient descent

similarly as in the continuous case is dangerous when using discrete variables. We validate that the estimation of $g(x^{(ik)}) - g(x)$ using $\tilde{d}_{ik}$ is sensible by computing the correlation for some elements of each domain between those two quantities. For instance, for the discrete LSI experiment, we find an average correlation $\rho = 0.55$, which means that even though the step between $x^{(ik)}$ and $x$ might be large, gradient information is indeed relevant. We give more details about this analysis in Appendix E.

We display some samples of the final repertoire obtained by ME-GIDE on Figure 3. To visualize the 3D structure of those proteins, we use AlphaFold2 Jumper et al. (2021b). One can see that ME-GIDE manages to extract proteins with various secondary and tertiary structures and that those proteins have high confidence score according to AlphaFold. We also conduct further validation on the protein experiment. We subsample our repertoire by recomputing larger cells to obtain a repertoire with 300 cells. Then, we insert the original repertoire into this smaller one to obtain the most fit solutions in each region. We firstly analyze the diversity in the sequence space, defined as the average edit distance between obtained solutions. We find

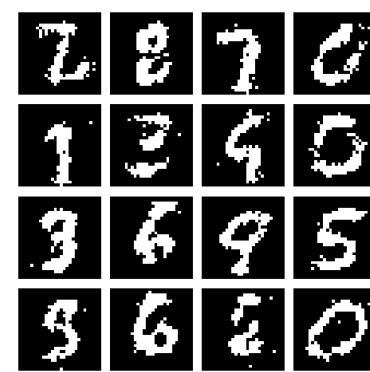

Figure 4: Samples from ME-GIDE final repertoire on the Binarized Digits experiment. We downsample the $40,000$ elemnts in the reprtoire by clustering its centroids to $16$ and display the element in each cell.

that ME-GIDE reaches an average distance of $106.9$ whereas MAP-ELITES gets $97.2$. We also use *S4Pred* Moffat & Jones (2021) to predict the secondary structure of each protein and use the average edit distance between secondary structures. ME-GIDE outperforms MAP-ELITES, obtaining an average distance of $44.2$ against $34.9$, and finds more diverse structures while also reaching an overall higher average fitness.

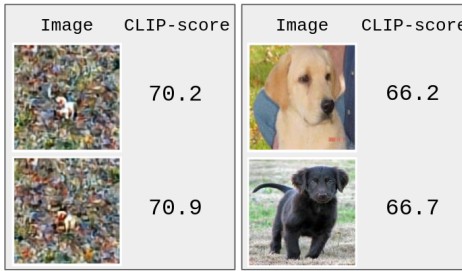

Figure 5: Samples found by ME-GIDE on the "Labrador" prompt against highest scoring images from ImageNet dataset. While the images found by ME-GIDE are not natural images, they achieve a higher score to be a "labrador" according to CLIP.

On Figure 4, we first display some samples found by ME-GIDE on the binarized MNIST experiment. One can see that ME-GIDE is able to find diverse images that resemble MNIST images and covers every digit. We also visualize on Figure 5 top-performing images from the repertoire learned by ME-GIDE as well as natural images from the ImageNet dataset. Images found by ME-GIDE cannot be considered to match the prompt from a human point of view although a small shape of a dog appears in the center of the image. It should still be noted that these images have higher CLIP scores than natural images that humans would score as perfectly aligned with the score prompt. In a sense, ME-GIDE manages to "fool" CLIP. Our work emulates the findings of Nguyen et al. (2015a) where Evolutionary Algorithms manage to "fool" a classifier . Our study highlights the difficulty of directly sampling relevant latent codes in the latent space of a discrete VAE, which is in general bypassed by an additional auto-regressive model that learns to sample meaningful latent codes.

# 5 RELATED WORKS

**Discrete Optimization** Discrete optimization Parker & Rardin (2014) is a longstanding branch of optimization where some of the variables or constraints are discrete. It is also named combinatorial optimization as the set of feasible solutions is generally finite but evaluation every solution is irrealizable. A first class of algorithms search for an exact solution: in particular, Integer Linear Prgoramming Schrijver (1998) has been the main subject of interest with exact methods such as cutting plan Gilmore & Gomory (1961) or branch and bound Land & Doig (2010). On the other hand, evolutionary algorithms have been very popular as they allow for more flexibility and can be stopped at any iteration to get the best solution explored so far Dorigo et al. (1999). Most of those

algorithms either incorporate this knowledge by mutating only to feasible solutions or use a relaxation and then project back to the feasible set Lin & Hajela (1992). For instance, for protein design, exact methods have been proposed for specific biophysical cost functions related to the potential energy Desmet et al. (1992) of the protein or its binding affinity Ojewole et al. (2018). However, due to the limitations in flexibility of the previous methods, the most popular design tool, incorporated in the popular *Rosetta* library, is based on simulated annealing Huang et al. (2011). Several work tried to leverage recent breakthroughs of deep learning methods tailored to proteins in the protein design, such as *AlphaDesign* Jendrusch et al. (2021) or *TrDesign* Norn et al. (2021). To the best of our knowledge, our work is the first to propose QD as a solution for protein design.

**Quality Diversity Optimization** Searching for novelty instead of quality Lehman & Stanley (2011a) was the first work formulating diversity as an end in itself. It was further refined by introducing a notion of local quality with local competition Lehman & Stanley (2011b). MAP-ELITES Mouret & Clune (2015) refined the notion of diversity by introducing the notion of repertoire over a set of descriptors. Further improvement were made on the design the descriptor space such as using a Voronoi tessellation Vassiliades et al. (2017) or use unsupervised descriptors Cully (2019); Grillotti & Cully (2022) as defining tailored descriptors can be tedious for some tasks. More efficient ways to illuminate this space have been proposed, such as more efficient mutation operators Vassiliades & Mouret (2018) or covariance matrix adaptation Fontaine et al. (2020). Closest to this work is Differentiable QD Fontaine & Nikolaidis (2021) where gradients are used to directly update over the continuous variables.

Application specific methods were developed to apply QD algorihtms to noisy domains Flageat & Cully (2020) or when multiple objectives are at stake Pierrot et al. (2022b). Reinforcement Learning is one of the most popular application of QD Arulkumaran et al. (2019) and some methods try to incorporate diversity directly in RL algorithms Nilsson & Cully (2021); Pierrot et al. (2022a). QD methods have already been applied to Latent Space Exploration, one of the domains we experiment on. The earliest work was named Innovation Engines Nguyen et al. (2015b) where authors try to generate diverse images using a QD approach over ImageNet. Later Latent Space Illumination has been introduced to generate game levels Schrum et al. (2020) or images Fontaine & Nikolaidis (2021). Our work is also applicable to LSI in the case of a discrete latent space and is the first work of its kind to the best of authors' knowledge.

# 6 CONCLUSION

We introduced ME-GIDE, a genetic algorithm that leverages gradient information over discrete variables for Quality Diversity optimization and show it outperforms classical baselines. Our entropy-based guidance allows to ease the search for good hyperparameters which can be tedious for some applications. Our experiment on protein design is the first using QD methods to the best of author's knowledge and this opens a way for more applications in the future. Indeed, we believe that generating not only fit but also diverse proteins can be useful as the fitness given by the model does not always translate to good properties in the lab.

Our goal was not to propose a novel image generation technique but while the results of our LSI experiment are interpretable and ME-GIDE finds images with higher fitnesses than realistic images, further research can be done to better understand and exploit these results. Our method can be used to create adversarial examples for generative models using discrete latent spaces, that are becoming more and more popular. On the other hand, to generate more realistic images, we could constrain the search to a few latent codes and use a recurrent architecture to generate a more realistic prior, at the cost of potentially facing vanishing or exploding gradients. We leave these open questions for future works.

As our method involves computing gradients of the fitness and descriptor functions, we expect it to be intractable for functions involving large neural networks, such as AlphaFold. Using a surrogate model to guide the search while keeping a large model for evaluation or validation to handle those cases could be a good research direction.

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
