# OpenReview forum: "Gradient-Informed Quality Diversity for the Illumination of Discrete Spaces"
_ICLR.cc/2023/Conference — Submitted to ICLR 2023_

### Official Review · Reviewer_uBn2 · 2022-10-16

**Confidence:** 4
**Correctness:** 2
**Technical Novelty And Significance:** 1
**Empirical Novelty And Significance:** 1
**Recommendation:** 1

**Clarity, Quality, Novelty And Reproducibility:**

Overall I don't think the proposed technical idea is novel: using gradient information to approximate a function locally during function optimization has been used in numerous research works, and sampling according to Boltzmann distribution (Eqn. 4) has also been commonly used in previous works.

The experiments are basically unreadable. I will give a few concrete examples listed below:

1. For the particular applications (protein folding, binarized MNIST, LSI), the authors does not even provide clear definition of the criteria (e.g., what is the meaning of "diverse patterns" in Fig. 3?) While the notion of the key metric "QD score" appears everywhere, I cannot find its formal definition, nor its connection to real-world significance. So why do we care about the score?

2. There are many undefined concepts and vague sentences in the experiment section. For example, Fig. 2 is hard to understand. What is ME? Is ME the same as MAP-ELITES? From the middle/bottom row of Fig. 2, I don't even know what to look at in these figures to understand the claim that the proposed method is better than the baseline.  In Sec. 4.2, "finds more diverse structures while also reaching an overall higher average fitness." -> How much higher the fitness becomes? Fig. 2 only shows QD score. Overall the experiments have no clear definition of which metric to look at and no concrete numbers to compare against, both for baselines and for proposed methods.

3. There is no obvious baselines as well as ablation studies about their proposed method. How is the method compared to random search techniques such as ARS (https://arxiv.org/abs/1803.07055), or evolutionary algorithm such as CMA-ES (https://arxiv.org/abs/1604.00772) plus diversity enhancement as used in MAP-ELITES? While both of them are methods in the continuous domain, they can be used in the discrete domain with the modification proposed in this paper. In Sec. 4.1, what pre-trained VQ-VAE model do you use? If you use a customized trained model, please report its performance to verify its usage.

**Strength And Weaknesses:**

Strength
1. The paper is straightforward

Weakness
1. The core idea (sampling nearby solutions according to first-order local function approximation guided by gradient) has been used in numerous works and is not novel at all.
2. The description of experiments is largely unclear, the significance of the results are unclear.

**Summary Of The Paper:**

This paper proposes ME-GIDE, an algorithm that aims to find a diverse set of solutions to a black-box function, which is referred as "Quality Diversity (QD)" in the paper. ME-GIDE extends existing MAP-ELITES algorithm, which is a genetic algorithm, to handle discrete inputs, by leveraging the gradient information of the function w.r.t the continuous inputs. The extension, known as Gradient Informed Discrete Emitter (GIDE), is straightforward: sampling neighboring solutions of the current discrete solution with the probability (Eqn. 4) that is proportional to the first-order function approximation by gradient at the current solution (Eqn. 2). The evaluation is done in protein folding domains.

**Summary Of The Review:**

Overall the paper requires substantial work and I vote for rejection.

---

> ### Author Response · Authors · 2022-11-18
> **Response to reviewer uBn2**
>
> * “The core idea (sampling nearby solutions according to first-order local function approximation guided by gradient) has been used in numerous works and is not novel at all.”
>
> We do not claim that using gradients to perform first-order approximation is novel. However, we would wholeheartedly welcome references of other works that apply such methods to discrete optimization and quality-diversity optimization and compare our contribution and results with those.
>
> * “For the particular applications (protein folding, binarized MNIST, LSI), the authors does not even provide clear definition of the criteria (e.g., what is the meaning of "diverse patterns" in Fig. 3?)”
>
> We define the criteria for each solution in the experiment section. To facilitate the reading, we will add a table to make those explicit in future revisions. The specific case of “diverse patterns” in Fig. 3 is further analyzed by an analysis on primary and secondary structures with more details in Appendix F. This analysis was performed as an additional validation as the diversity criterion in the protein experiment is defined in an unsupervised manner. We validate this way of defining the descriptor space by showing that this translates into actual biological diversity in the obtained proteins.
>
> * “There is no obvious baselines as well as ablation studies about their proposed method.”
>
> We agree that there are no obvious baselines for the problem of Quality-Diversity Optimization for Discrete Spaces, which is why we adapted MAP-Elites to handle discrete data. ARS could indeed be adapted to discrete data but does not handle diversity. Adapting CMA-ES to discrete data would be a challenging task in itself as it samples candidates using a normal distribution. We considered CMA MAP-Elites (Fontaine et al., 2020), which is tailored for Quality-Diversity Optimization but suffers from the same problem.
>
> * “In Sec. 4.1, what pre-trained VQ-VAE model do you use? If you use a customized trained model, please report its performance to verify its usage.”
>
> We followed the guidelines of the paper that introduced VQ-VAE (Van Den Oord et al., 2017) and give details in Appendix C.
>
> [Van den Oord et al., 2017] Van Den Oord, Aaron, and Oriol Vinyals. "Neural discrete representation learning." Advances in neural information processing systems 30 (2017).
>
> [Fontaine et al., 2020] Fontaine, Matthew C., et al. "Covariance matrix adaptation for the rapid illumination of behavior space." Proceedings of the 2020 genetic and evolutionary computation conference. 2020.

---

### Official Review · Reviewer_MZFX · 2022-10-20

**Confidence:** 3
**Correctness:** 4
**Technical Novelty And Significance:** 2
**Empirical Novelty And Significance:** 2
**Recommendation:** 6

**Clarity, Quality, Novelty And Reproducibility:**

Clarity:

The paper is clearly written and the approach is well described.

Quality:

The paper presents results on some challenging, high-dimensional problems. I am not surprised that the gradient-based approach improves performance. But still, the results are compelling and I believe this could be a useful tool for practitioners who are using this kind of optimization algorithm.

Novelty:

This work applies and existing, successful idea to an existing successful algorithm. I do not believe this to be the most novel work in the world but, as it stands, I have not yet seen these ideas applied towards optimization. So, it is nice to observe that these ideas which have been very successful for sampling, can also be applied to optimization.

Reproducibility:

The experimental details are well documented and the method is simple and well-described so I am fairly confident the results are reproducible.

**Strength And Weaknesses:**

Strengths:

The paper is clearly written and does a good job to introduce a the reader to the problem of quality diversity optimization (of which I am not an expert). The method is simple and well motivated for the problem at hand and is simple to implement and apply. The results are compelling (in particular the protein results).

Weaknesses:

While the presented empirical results are compelling, one is left to wonder when this method will fail. Recent works on discrete MCMC have presented some theory on when those methods will work well and when they won't. These arguments are typically made based on the Lipschitz constant of the model's log-likelihood function. I assume a similar argument could be applied here but I am not as familiar with the nuances of the quality diversity optimization problem to know that for sure. I feel the paper would be greatly strengthened with some (minor) theoretical justification on the settings where this approach is likely to work and when it is not.

As a non-expert in this exact domain, I left this paper wondering how much the choice of local regions effects the resulting solution. I feel the paper would be strengthened with some discussion of how this should be chosen to optimize performance.

**Summary Of The Paper:**

This paper addresses the problem of quality diversity optimization. In short, this problem is to find a set of which maximize some function f(x) where each x in this set is the optimal value in some region of the space of possible x's. The space where these regions are define is typically lower dimensional than the space of x and is chosen by the user.

This problem has a number of successful solutions in continuous spaces. In discrete spaces, solutions exist but most rely on random search. This paper builds on recent developments in discrete MCMC sampling which notice that many discrete problems are often naturally embedded in continuous spaces and represented by continuous, differentiable function. In such settings the gradients of these functions are often useful for guiding discrete search. This paper applies this idea to quality diversity optimization in discrete spaces through a new local search proposal distribution.

This new proposal plugs in naturally to existing approaches and leads to improved performance over random search and an alternative gradient-based method. The authors demonstrate this on a diverse set of tasks ranging from protein design to image generation.

**Summary Of The Review:**

This work applies recent successful advances from discrete sampling to quality diversity optimization in discrete spaces. The new method plugs in easily with existing approaches and performs well on a diverse set of challenging discrete problems. The method is not particularly novel but the results are compelling and the method is neat and simple.

---

> ### Author Response · Authors · 2022-11-18
> **Response to Reviewer MZFX**
>
> * “I feel the paper would be greatly strengthened with some (minor) theoretical justification on the settings where this approach is likely to work and when it is not. ”
>
> We concur with your remark. However, we would like to highlight that convergence guarantees of population based optimization is an active yet difficult area of research. Our approach is oriented towards experimentations to leverage the successes of population based methods to the optimization of  high dimensional complex objective function.
>
> * “As a non-expert in this exact domain, I left this paper wondering how much the choice of local regions affects the resulting solution.”
>
> The design of descriptors and the grid have a strong effect on the ability of finding good solutions. Namely, the grid must contain a sufficient number of cells to help exploration. In our experiments, we chose not to handcraft those descriptors and grid by using unsupervised methods to design them (Cully, 2019).
>
> [Cully, 2019] Cully, Antoine. "Autonomous skill discovery with quality-diversity and unsupervised descriptors." Proceedings of the Genetic and Evolutionary Computation Conference. 2019.

---

### Official Review · Reviewer_i5mR · 2022-10-24

**Confidence:** 4
**Correctness:** 2
**Technical Novelty And Significance:** 2
**Empirical Novelty And Significance:** 2
**Recommendation:** 1

**Clarity, Quality, Novelty And Reproducibility:**

Besides the typos, there are many points that are unclear, which make it impossible to thoroughly evaluate the submission.  These include the following.

Page 5: The discussion on adjusting T is vague.  Why those specific trial values of H_{target}?  What is the function that you are optimizing via GD?

Where are the appendices that you refer to?   They are not in the pdf I downloaded.  (I found them buried in a supplemental link, but they should be part of the pdf.  Also, they do not completely address my comments in this section.)

In the caption of Figure 3, explain what the reader should pay attention to to notice "diverse patterns in the secondary structure".

On page 7, you say that you report results for "the best set of hyperparameters".  Best on what data set?  If you mean the best on an independent validation set (or via cross-validation), then that is probably okay if there is another separate test set.  However, if what you report are the best results on the final test set, then your results are cherry-picked.

Page 8: Explain the assertion that ME-GIDE finds various structures with high confidence scores according to AlphaFold.  What were those scores?  Were they better than those found by other methods (especially by non-GIDE methods)?

The authors define the problem in terms of f(.) and c_i(.), but never specify what these are in your experiments. They also do not define "QD score" in Figure 2.  These are glaring omissions.

The bottom two rows of Figure 2 (describing the repertoires) are not explained, so it is difficult to interpret these meaningfully.


**Strength And Weaknesses:**

Strengths:
- Extending gradient-based search to discrete spaces is an important problem.

Weaknesses:
- The paper is sorely lacking in many details (listed in the "clarity" section).
- Several typos, including but not limited to:
  - Abstract: Need a comma after "However".  Also at other instances of this word, including first line of the second paragraph of Section 1.
  - Page 2: "gradient informed" -> "gradient-informed"
  - Page 2: "algorithm, discretize" -> "algorithm discretize"
  - Algorithm 1: "Adjust T" -> "Adjust temperature parameter T"
  - Page 3: "called archive" -> "called an archive"
  - Page 3 and elsewhere: Change all instances of "firstly" to "first", "secondly" to "second", etc.
  - Page 3 and elsewhere: Add a comma or semicolon at the end of each item in an enumerated list, e.g., "repertoire (ii)" -> "repertoire; (ii)"
  - Page 3: Capitalize "gaussian".
  - Page 3: "(Fontaine & Nikolaidis 2021)" -> "Fontaine & Nikolaidis (2021)"
  - Page 3: Place a comma after "(MEGA)"
  - Page 3: The period after "g(z)-g(x)" should be a comma.
  - Page 3: You use H for entropy and a Hamming ball.  Change one of these for clarity.
  - Page 4: Put a comma after "Formally".
  - Page 5: "human-interpretability" -> "human interpretability"
  - Page 5: What is a "degenerated" distribution?
  - Page 5 and elsewhere: Put a comma after "i.e." and "e.g."
  - Page 5: Put "T" in math mode to italicize
  - Page 5: "amino-acids" -> "amino acids"
  - Page 5 and elsewhere: Use proper LaTeX quotes, i.e., two back quotes to open and two single quotes to close.  E.g., ``Rat sarcoma virus'' instead of "Rat sarcoma virus"
  - Page 5: Add a space in ".Following"
  - Page 5 and elsewhere: Add a comma after "Finally"
  - Page 6: Use math mode for MNIST sizes, i.e., "28x28" -> "$28 \times 28$"
  - Page 6: "uses" -> "use" after "Fontaine & Nikolaidis (2021)"
  - Page 7: "fewer iteration" -> "fewer iterations"
  - Page 7: "conclusion are" -> "conclusions are"
  - Page 9: "popular application" -> "popular applications"


**Summary Of The Paper:**

To address the quality diversity problem in discrete search spaces, the authors modify the existing algorithm MAP-ELITES to employ gradient information, yielding the ME-GIDE evolutionary algorithm.  They evaluate their method on protein data, MNIST, and ImageNet.


**Summary Of The Review:**

The paper is far too vague in too many ways to merit publication.

---

> ### Author Response · Authors · 2022-11-18
> **Response to Reviewer i5mR**
>
> * “The discussion on adjusting T is vague. Why those specific trial values of H_{target}?”
>
> Our method is a heuristic and is hence dependent on hyperparameters. The goal of introducing the entropy is to give an intuitive understanding of how to adjust T. Fixing H_{target}=0 always promotes going in the direction of the highest gradient, while fixing  H_{target}=1 gives a uniform distribution at sampling time which is equivalent to standard MAP-Elites.
> We tried those specific values of 0.4, 0.6 and 0.8 with other values as well. We found out that those values were the best performing overall and this H_{target} reduces the search space for hyperparameters. We also want to highlight that in reinforcement learning a control of the entropy is quite common.
>
> * “Where are the appendices that you refer to? They are not in the pdf I downloaded.”
>
> We followed the authors’ guidelines and split the pdf in one main paper and appendices to attach to the supplementary materials, as is standard in most conferences.
>
> * “On page 7, you say that you report results for "the best set of hyperparameters". Best on what data set? If you mean the best on an independent validation set (or via cross-validation), then that is probably okay if there is another separate test set. However, if what you report are the best results on the final test set, then your results are cherry-picked."
>
>
> As the baselines we use are heuristics, their results are dependent on the choices of hyperparameters. To make the comparison as fair as possible, we chose to report the best result for the baselines on the test dataset. Indeed, we cherry-picked the results in favour of the baselines (but not for ME-GIDE) and ME-GIDE consistently outperforms them without cherry-picking as outlined in Appendix B.
>
> * “Page 8: Explain the assertion that ME-GIDE finds various structures with high confidence scores according to AlphaFold. What were those scores? Were they better than those found by other methods (especially by non-GIDE methods)?”
>
> In terms of diversity, we conduct an analysis on first and secondary structure using S4Pred (Moffat et al., 2021) diversities, where the scores are reported and more details are provided in Appendix F.
>
> * "The authors define the problem in terms of f(.) and c_i(.), but never specify what these are in your experiments. They also do not define "QD score" in Figure 2. These are glaring omissions."
>
> We will add a table defining clearly those terms in future revisions. The QD score is defined in Equation (1) on the second page of the paper.
> The bottom two rows of Figure 2 (describing the repertoires) are not explained, so it is difficult to interpret these meaningfully.
>
> Bottom two rows are common representations in quality diversity optimization literature of the explored space of solutions.
> Practically the value of each pixel is a local maximum.
>
> [Moffat et al., 2021] Moffat, Lewis, and David T. Jones. "Increasing the accuracy of single sequence prediction methods using a deep semi-supervised learning framework." Bioinformatics 37.21 (2021): 3744-3751.

---

### Official Review · Reviewer_xfZC · 2022-10-31

**Confidence:** 2
**Correctness:** 3
**Technical Novelty And Significance:** 3
**Empirical Novelty And Significance:** 2
**Recommendation:** 6

**Clarity, Quality, Novelty And Reproducibility:**

plz see my comments above, I think experiment section will need some improvement.

Questions:
1) why you use equation 3 to encourage exploration? What's the intuition behind of it? Thank you.

**Strength And Weaknesses:**

Strength:
1) the algorithm seems to be a straightforward extension of an existing algorithm framework to make it work for a specific domain.
2) the authors evaluate the algorithm on many interesting datasets and tasks, and show sizable improvements on these tasks

Weakness:
1) the paper needs to be re-organized in experiments. For example, a table with the performance of all baseline methods clearly listed will make the reviewers life much easier, than than going into your text to dig these results out ;)
2) The baseline in experiments are very limited, it seems the author only picks one baseline for each experiment setup. Please correct me if I'm wrong. Please also clarify if there is only 1 baseline exists (which I don't think to be true)
    a) protein: ME-GIDE V.S. ME-Elite
    b) ImageNet: ME-GIDE V.S. ImageNet
3) I also don't get the point of Fig.5. The image generated from ME-GIDE is not intuitive to human but with higher CLIP-score. I believe there are other metrics to be used in together with CLIP in image generation. How are other metrics like? Why ME-GIDE generates some non-sense images to human? I understood your point is using CLIP to measure the diversity. But this seems to me the ME-GIDE lacks a some protection mechanism to guarantee the correctness of optimization.


**Summary Of The Paper:**

This paper presents an interesting extension of map-elite algorithm, ME-GIDE, to address the quality-diversity optimization with discrete inputs but differential objective and description functions. The authors evaluate their algorithm on several challenging benchmarks, including protein designs and image generation.

**Summary Of The Review:**

I'm not an experts in this area, but I found interesting piece in this paper. However, this paper definitely needs some improvement either in writing or experiments.

---

> ### Author Response · Authors · 2022-11-18
> **Response to Reviewer xfZC**
>
> * “the paper needs to be re-organized in experiments”
>
> We thank you for your remark and concur with your conclusion to reorganize the experiments to highlight the benefits of the presented method.
>
> * “The baseline in experiments are very limited, it seems the author only picks one baseline for each experiment setup”
>
> To the best of our knowledge, the problem of Quality-Diversity Optimization for Discrete Data has not been extensively studied before, which explains the limitations in the number of alternative baselines we could try. To clarify, here, we compare our own GIDE emitter with two main baselines: MAP-Elites with random mutations (ME) and MAP-Elites with a pseudo gradient descent where after each gradient step, the candidate is projected back to the discrete space (ME-GDP). Those two baselines are kept the same in each experiment.
>
> * “I also don't get the point of Fig.5. The image generated from ME-GIDE is not intuitive to human but with higher CLIP-score.”
>
> We want to specify again that our method does not aim to compete with other image generation methods. The setting of this experiment is inspired by previous works where authors use genetic algorithms to generate images by maximizing the “score” given by a neural network such as a pre-trained classifier on ImageNet (A. Nguyen et al., 2015) or a StyleGAN (M. Fontaine et al., 2021). What we find out is that CLIP can easily be fooled in the sense that it is possible to generate images with a higher CLIP score towards the prompt “a labrador” than natural images of labradors. This result is in line with other works on classifiers (Nguyen et al., 2014).
>
> * “But this seems to me the ME-GIDE lacks a some protection mechanism to guarantee the correctness of optimization.”
>
> We believe on the contrary that ME-GIDE is performing the optimization task well and does find better solutions than alternative methods towards the CLIP score. Indeed, in order to generate natural images, it would be preferable to modify the way images are scored to penalize un-natural images.
>
> * “why you use equation 3 to encourage exploration? What's the intuition behind of it?"
>
> Equation 3 describes the sampling mechanism of our emitter. As our method makes use of gradients, we could have chosen to mutate the candidate to the neighbor with the highest d. However, exploiting the highest values in the scoring function may lead to minimal exploration and for instance getting stuck in a local optimum. In contrast sampling according to d with an entropy constraint aims at arbitrating exploitation and exploration
>
> [A. Nguyen et al., 2015] Nguyen, Anh Mai, Jason Yosinski, and Jeff Clune. "Innovation engines: Automated creativity and improved stochastic optimization via deep learning." Proceedings of the 2015 Annual Conference on Genetic and Evolutionary Computation. 2015.
>
> [M. Fontaine et al., 2021] Fontaine, Matthew, and Stefanos Nikolaidis. "Differentiable quality diversity." Advances in Neural Information Processing Systems 34 (2021): 10040-10052.

---

### Author Response · Authors · 2022-11-18
**General response to reviewers**

We thank the reviewers for their comments that we will take into account when revising the paper. We answer each reviewer independently point by point.

---

### Decision · Program_Chairs · 2023-01-20

**Decision:**

Reject

**Justification For Why Not Higher Score:**


As most reviewers commented,

- The main idea of the method is relatively straightforward and not novel.

- The experiment part is weak.

**Justification For Why Not Lower Score:**

N/A

**Metareview: Summary, Strengths And Weaknesses:**


In this paper, the authors consider the quality diversity (QD) problem, which aims for finding a diverse set of solutions over discrete variables. The authors propose ME-GIDE that exploits the gradient approximation to reduce the function calls, and test the algorithm on a variety of problems.

The method section is well-organized and easy to follow.

The major concerns raised by most of the reviewers lie in two-fold:

-  the idea is straightforward, considering the existing successful discrete sampling algorithms.

- the competitors in experiment section are limited, and the empirical results are not presented well.

Reviewer MZFX also suggested to explore the theoretical justification of the algorithm.

Please consider the suggestions provided by the reviewers to improve the draft.